# Quality of life in hypertensive patients using the WHOQOL-BREF instrument in the post-pandemic Bangladesh: A cross-sectional study

Nasrin Akter[1]*, Farhana Faruque Zerin[1], Bilkis Banu[2], Fatema Afrin Kanta[1], Shahnaz Begam[3], Sarder Mahmud Hossain[1]

1 Department of Public Health, Northern University Bangladesh, Dhaka, Bangladesh, 2 Independent University, Bangladesh (IUB), Dhaka, Bangladesh, 3 SAIC College of Medical Science & Technology, Dhaka, Bangladesh

* nasrin.ddc@gmail.com

## Abstract

### Background

To combat growing prevalence of hypertension in Bangladesh, it is critical to have an in-depth understanding about quality of life (QOL) among people living with hypertension and related factors. In the recent COVID-19 pandemic the QOL of hypertensive people got downsized. This study aimed to measure QOL among hypertensive people in a selected tertiary hospital in Dhaka city, and its association with the basic characteristics of the patients.

### Methods

This study was conducted among randomly selected 300 hypertensive patients from two departments of Square Hospitals Limited, using the patient register record. Data were collected through face-to-face interview methods. The WHOQOL-BREF questionnaire was used to assess the QOL of the subjects. Descriptive statistics were used to examine mean scores of quality of life. Cronbach's alpha coefficient and Pearson's correlation coefficient were applied to estimate the internal consistency, and the level of agreement among different domains of WHOQOL-BREF, respectively. Chi-square test followed by binary regression analysis was used to measure the association between QOL domains and independent variables.

### Results

Both overall WHOQOL-BREF and each domain had a good internal consistency, (r = 0.13–0.77, p < 0.01). The QOL among hypertensive patients was found poor in psychological (71%) and social (74.7%) domain and good in environment (63%) and physical (65%) domain. Backward binary regressions revealed that being older over

**Data availability statement:** All relevant data are within the manuscript and its Supporting information files.

**Funding:** The author(s) received no specific funding for this work.

**Competing interests:** The authors have declared that no competing interests exist.

55 years (p = 0.01), diabetic (p = 0.02), having history of COVID-19 (p = 0.01) and poor monthly family income (USD ≤ 853.14) (p = 0.01) were significantly associated with poor QOL in all domains. Moreover, older age (p = 0.01) and poor lifestyle (p = 0.02) were significantly associated with poor overall quality of life and poor perception of general health.

## Conclusion

The results revealed low QOL in psychological and social domain, including significant factors associated with the poor QOL in all domains. Planning and implementation of effective interventions are needed to improve QOL among hypertensive patients targeted towards aged, diabetic, lower income group who had positive COVID-19 infection and poor lifestyle through health system strengthening.

## 1 Background

Hypertension, commonly referred to as high blood pressure, is one of the leading causes of death among non-communicable diseases worldwide. It places a significant pressure on healthcare systems and affects millions of people, ultimately impacting both quality of life and mortality rates [1]. Hypertension is often called the "silent killer," as it generally remains asymptomatic until it leads to severe health complications such as heart disease, kidney failure, or stroke [1]. According to estimates by the World Health Organization (WHO), approximately 1.28 billion adults globally suffer from hypertension, with a considerable portion residing in low- and middle-income countries like Bangladesh [2]. Although hypertension is preventable and manageable, it remains largely controlled. The global rate of uncontrolled hypertension declined slightly from 29% to 26% between 2010 and 2019, remaining far below WHO's target of a 25% reduction by 2025 [3]. In Bangladesh, nearly one in five adults is affected by hypertension, making it a pressing public health issue with lasting implications for both individuals and society [4].

The COVID-19 pandemic exacerbated the hypertension crisis by increasing stress levels, disrupting medical services, and discouraging people from seeking healthcare [5]. In Nepal, a study found that 27.8% of hypertensive patients experienced anxiety and 24.3% suffered from depression during the pandemic, highlighting the significant mental health burden in this group [5]. These findings emphasize the impact of pandemic-related stress and healthcare disruptions on hypertension rates in South Asia. Similarly, in Bangladesh, research indicated that the prevalence of self-reported hypertension among older adults rose significantly from 43.7% in 2020 to 56.3% in 2021, highlighting a notable increase in cases during the pandemic [6]. The psychological and social well-being of individuals with high blood pressure was deeply impacted due to prolonged isolation, economic difficulties, and limited access to healthcare services [6]. These challenges stress the importance of evaluating how the pandemic has influenced the quality of life for hypertensive individuals, particularly as society transitions into the post-pandemic period.

The WHO defines quality of life (QOL) as how people feel about their well-being in relation to their goals, expectations, and surroundings [7]. For people with hypertension, QOL often worsens due to the long-term nature of the disease, other health conditions, and the stress it brings [8]. WHOQOL-BREF is a well-known tool that measures QOL in four key areas: physical health, mental well-being, social relationships, and environment [9]. While this tool is widely used in high-income countries, it has mostly been applied to diabetes and stroke patients in low- and middle-income countries (LMICs) like Bangladesh [10–13]. In Bangladesh, researchers have used the health-related quality of life (HRQoL) tool to assess QOL in hypertensive patients [14]. In Indonesia, during the pandemic, QOL among hypertensive patients was evaluated using the European Quality of Life-5 Dimensions (EQ-5D) tool [15]. While tools such as the EQ-5D are widely used for assessing QOL, they primarily focus on health utility values and are more suitable for economic evaluations. In contrast, WHOQOL-BREF offers a broader, multidimensional assessment—including physical, psychological, social, and environmental domains—making it more appropriate for understanding the lived experiences of hypertensive patients in a culturally nuanced context. Additionally, the WHOQOL-BREF has been cross-culturally validated and shown to be adaptable for use in LMICs like Bangladesh, supporting its relevance for this study. These findings reinforce the psychometric soundness of the WHOQOL-BREF for use in diverse Bangladeshi populations. However, to inform more comprehensive and culturally grounded healthcare strategies in Bangladesh, we considered the relative strengths of QOL assessment tools. Expanding the use of WHOQOL-BREF in these regions could provide a clearer picture of how hypertension impacts daily life and help improve healthcare strategies. While QOL research among chronic illnesses exists, hypertension warrants especial attention due to its asymptomatic nature and long-term behavioral management demands. Unlike conditions that present with immediate symptoms (e.g., diabetes or stroke), hypertension's "silent" progression can result in patients underestimating its seriousness, thereby diminishing adherence to treatment and engagement with health services. Moreover, the post-pandemic psychosocial stressors—such as social isolation, fear of comorbid risk, and disrupted routines—may uniquely impact individuals with hypertension. Thus, generalized QOL findings from other illness groups may not capture the nuanced experiences of hypertensive patients in Bangladesh. This study specifically addresses this gap by exploring their QOL through a multidimensional and culturally adaptable tool.

Research shows that when people with high blood pressure have a lower quality of life, they are less likely to stick to their treatment, which leads to higher medical costs and worse health outcomes [16–19]. The COVID-19 pandemic made this even more challenging by reducing physical activity, increasing stress, and limiting access to healthcare [14,15]. Despite these issues, there is very little research in Bangladesh on how hypertension affects quality of life, especially using reliable tools like the WHOQOL-BREF during or after the pandemic. Without this data, it's hard to develop effective and culturally appropriate strategies to support and improve the well-being of hypertensive patients.

The growing number of hypertension cases in Bangladesh is seriously affecting people's quality of life, highlighting the need for stronger public health efforts. The COVID-19 pandemic has made things even more challenging, making it crucial to assess how patients are coping after the pandemic. Using a reliable tool like WHOQOL-BREF can offer valuable insights tailored to the local context, helping to develop better healthcare strategies and improve the lives of those with hypertension. This approach can provide culturally relevant insights to guide effective healthcare interventions and improve the well-being of individuals living with hypertension. Addressing this gap, the present study aims to evaluate the QOL of hypertensive patients in a post-pandemic Bangladeshi setting using the WHOQOL-BREF instrument. By focusing on Square Hospitals Limited, a leading tertiary care facility in Dhaka, the study explores how quality of life (QOL) is influenced by various sociodemographic, lifestyle, and clinical factors among hypertensive patients. The findings will provide valuable insights into what affects the quality of life (QOL) among hypertensive individuals, guiding resource allocation and intervention strategies tailored to local needs. Furthermore, this research seeks to provide actionable insights into integrating QOL assessments into hypertension care, thereby addressing broader public health goals.

## 2 Methods

### 2.1 Study design

A quantitative cross-sectional study with an analytical approach was conducted from 01 March 2024–30 August 2024, as data were collected from a group of subjects at a single point in time and data evaluation needed to be done critically. Semi-structured data were collected for obtaining the information on socio-demographic, clinical characteristics, and quality of life using the WHOQOL-BREF instrument among hypertensive patients of Dhaka district of Bangladesh.

### 2.2 Study participants, sampling and sample size

This study included a total of 300 hypertensive patients admitted to Square Hospitals Limited, located in West Panthapath, Kalabagan, Dhaka, Bangladesh. The quantitative information for this study was collected from patients who met the following criteria: they had hypertension, were suffering from coronary artery disease, were admitted under the departments of Cardiac & Vascular Surgery and Cardiology (Interventional) in the selected hospital, were free from mental or physical disabilities and provided their consent to participate in this study. Participants had been diagnosed with hypertension for a minimum duration of six months prior to the survey, based on hospital records.

A random sampling technique was employed in this study to ensure the generalizability of the findings. Hypertensive patients from the Dhaka division were considered as the study population, among the eight divisions (Dhaka, Chittagong, Rajshahi, Khulna, Rangpur, Mymensingh, Sylhet and Barisal) of Bangladesh. Dhaka division was chosen as the study site through multi-stage random sampling and this division offers a dense population and a diverse patient base with extensive availability of healthcare services in the main city. The overall prevalence of hypertensive patients in the Dhaka reflected 31% [20]. Furthermore, the study was conducted at a randomly selected hospital (Square Hospitals Limited) from the top 10 tertiary care hospitals that have dedicated cardiac department [21]. These hospitals were ranked based on publicly available data from health service directories and included Square Hospitals Ltd., United Hospital Ltd., Evercare Hospital Dhaka, Apollo Imperial Hospitals, Labaid Specialized Hospital, Islami Bank Central Hospital, Central Hospital Ltd., Green Life Medical College Hospital, Popular Medical College Hospital, and Asgar Ali Hospital. The ranking considered criteria such as accreditation, service quality, cardiac specialization, patient satisfaction, and availability of multidisciplinary units. Square Hospitals Ltd. was chosen through simple random sampling from this list. Square Hospital is one of the largest 500-bedded general hospitals providing quality health care services in Dhaka, Bangladesh [22]. Initially, 600 hypertensive inpatients were identified from the admission registry of the Cardiac & Vascular Surgery and Cardiology (Interventional) departments. From this pool, every second patient was selected using systematic random sampling, resulting in a final sample of 300 participants. These departments were chosen due to their structured, well-documented care pathways for patients with hypertension and related cardiac conditions, facilitating reliable data collection. Patients managed in outpatient or general medicine settings were excluded due to variability in hypertension management protocols and inconsistent follow-up practices. The decision to focus on private hospitals was based on logistical and quality assurance factors: private hospitals in Dhaka maintain more consistent electronic health records, have established non-communicable disease programs, and allow for standardized patient follow-up.

Our study does include socioeconomic diversity over 65% of participants reported a monthly income of ≤ USD 853.14, suggesting the inclusion of middle- and lower-income urban patients not solely affluent individuals. Thus, it is allowing for partial mitigation of income-related selection bias. The initial sample size of 280 was calculated using the formula $n = Z^2 pq/d^2$, where $Z = 1.96$, $d = 0.05$, and $p = 0.76$. The value of p was based on the average of WHOQOL-BREF domain mean scores (range: 0.65–0.88) reported in a similar population [23], in the absence of precise binary prevalence data. Although this value originates from continuous data, it was used as a proxy to estimate the proportion of participants likely to report good quality of life. An additional 7% was added to account for potential non-response or error, resulting in a final sample size of 300.

## 2.3 Data collection

Quantitative data were collected from the hypertensive patients by using a pre-tested and semi-structured questionnaire through the face-to-face interview method. Pretesting was done among the 5% (15) hypertensive subjects of the total sample size who met the same inclusion criteria as the main study population, including diagnosis, age range, and a similar type hospital setting (National Heart Foundation, Dhaka). Then relevant modifications were done on the basis of the outcomes of the procedure to ensure the validity of the instrument. The WHOQOL-BREF was administered in Bengali, the native language of participants, ensuring cultural and linguistic appropriateness. Respondents were interviewed during the period of 20 March 2024–25 May 2024. Data collectors were well trained and closely monitored by the research team to avoid interviewer bias. Clinical parameters were obtained from patient records available in the hospital. Hypertension levels were categorized using JNC 8 criteria into Stage 1 (systolic 140–159 mmHg or diastolic 90–99 mmHg) and Stage 2 (systolic ≥160 mmHg or diastolic ≥100 mmHg) based on recent clinical measurements retrieved from patient charts. The interviewer took only 15–20 minutes to complete the survey. All authors had access to the collection and preserving participants' information during or after data collection. The survey was administered in the Bengali language with full support of the hospital administrative authority.

## 2.4 Questionnaire design

The questionnaire, derived from variables identified in published studies, was reviewed by two independent researchers and pre-tested on fifteen respondents. The feedback from the pre-test was utilized to improve the questionnaire. To gather comprehensive information, a semi-structured format was employed. The key components of the questionnaire were: (i) Socio-demographic information: gender, age, marital status, education, monthly family income; (ii) Anthropometric and Clinical characteristics: nutritional status by assessing Body Mass Index (BMI), fasting blood glucose, exposure to COVID-19; (iii) Pattern of lifestyle: physical activity (types & duration), sleeping pattern (duration, day time napping, insomnia) and deleterious habit (tobacco & alcohol consumption); and (iv) Quality of Life (QOL) and general health by using WHOQOL-BREF. To operationalize lifestyle patterns, three behavioral dimensions incorporating thirteen components were considered: (i) Food habit: Several types of food intake (healthy and unhealthy food) including amount of salt intake per day by the participants and their meal frequency were considered to assess their healthy or unhealthy dietary practice as part of their lifestyle pattern. More than 1 tea spoon salt intake was considered as unhealthy practice. In addition, alcohol intake was considered as unhealthy habit; (ii) Physical activity: Participants were considered physically active if they reported engaging in at least 30 or 150 minutes of moderate-intensity activity (e.g., walking) per day and week, in line with WHO guidelines [24]. Those reporting less than this threshold were classified as physically inactive or below standard practice; (iii) Sleeping pattern: Adequate sleep was defined as 6–8 hours (Standard sleeping) of nightly sleep without regular complaints of insomnia or no daytime napping. Respondents reporting less than 6 (below standard) or more than 8 hours of sleep (Oversleeping), or those with self-reported insomnia or frequent daytime napping, were considered to have a poor sleep pattern [25]; (iv) Perception of stress and anxiety: This included perception about stressed and anxious feeling. Participants who felt stressed and anxious were classified as having unhealthy lifestyle. Each of these components was scored as binary (0 = unhealthy; 1 = healthy). A cumulative score ranging from 0 to 13 was generated for each participant and then converted in percentage score. Those scoring ≤62% (mean 61.56) were classified as having a "poor lifestyle", while those scoring >62% categorized as having a "moderate/healthy lifestyle". This scoring approach was informed by previous studies assessing lifestyle risk aggregation in relation to chronic disease outcomes. WHOQOL-BREF was utilized in this study to assess the QOL among hypertensive patients. It was developed by the World Health Organization. This cross-cultural instrument captures a wide range of QOL aspects including physical health, psychological health, social relationship and environment. It comprises two items on Overall QOL and General Health, and 24 items of Satisfaction rated on a 5-point Likert scale. These 24 items are divided into four domains: Physical health with 7 items (DOM1),

Psychological health with 6 items (DOM2), Social relationships with 3 items (DOM3) and Environmental health with 8 items (DOM4). On a response scale, each item is scored from 1 to 5 in the WHOQOL-BREF [23].

## 2.5 Data analysis

The collected data were reviewed and analyzed employing the Statistical Package for the Social Sciences (SPSS) software, version 26.0. Descriptive statistics (frequency and proportions) were applied to summarize the study characteristics.

Continuous variables related to socio-demographic characteristics, such as age and monthly family income, were scored and categorized based on the mid-values of the percentage scores used as cut-off points [26]. Body Mass Index (BMI) was determined by dividing weight in kilograms by height in meters squared. The BMI thresholds were underweight (<18.5), normal range (18.5–24.9), and overweight (≥25). The threshold for fasting blood glucose (BG) was set at >7 mmol/L. As stated above, lifestyle was assessed using a thirteen-item index (fast food consumption, frequency of fish consumption, fruits & vegetable consumption per week, soft drinks consumption, frequency of meal intake per day, quantity of salt intake, alcohol consumption, Daily physical activity, weekly physical activity, sleeping duration, day time nap, perceived of stress, and anxious feelings), and participants were categorized into poor versus moderate/healthy lifestyle based on a cumulative score. The scores for the four domains (DOM 1, DOM 2, DOM 3, DOM 4), as well as overall perception of QOL (Q1) and general health (Q2) were divided into two groups considering the 50% as the cutoff point. Scores above 50% were considered as 'Good/higher QOL'; while below 50% as 'Poor to moderate/lower QOL' [27].

Variables were described using frequency, percentage, and measures of central tendency and dispersion, including the correlation coefficient. A multinomial logistic regression analysis was conducted, followed by a modeling procedure to adjust the confounders. This analysis utilized a backward elimination process, including pre-specified variables identified as significantly associated with the dependent variable by the chi-square test: gender, age, marital status, education, monthly family income, BMI, fasting blood glucose, exposure to COVID-19, pattern of lifestyle. Odds Ratios with 95% confidence intervals for the four domains, overall QOL, and general health (poor and good) were calculated for the specified exposures. Before performing binary logistic regression, relevant assumptions were checked. Multicollinearity was evaluated using Variance Inflation Factor (VIF), and all predictors showed VIF values below 2.0, indicating no multicollinearity. Independence of residuals was examined using the Durbin-Watson statistic, which fell within the acceptable range of 1.6 to 2.2. Additionally, the goodness-of-fit of the regression models was evaluated using the Hosmer–Lemeshow test, with p-values greater than 0.05 suggesting a good model fit. These checks confirmed the suitability of the data for regression analysis.

## 2.6 Ethical considerations

This study received approval from the Ethical Review Committee of the Department of Public Health at Northern University Bangladesh (NUB/DPH/EC/2024/34) and adhered to the Declaration of Helsinki guidelines. Participation was both anonymous and voluntary. Written informed consent was obtained from respondents at the start of the survey, with participants free to withdraw at any time, ensuring their autonomy was respected.

## 3 Results

### 3.1 Participant's characteristics

A total of 300 hypertensive patients were considered in this study with 59.7% female and a mean (±SD) age of 55.59 (±11.59) years. More than half of the respondents (50.3%, n = 151/300) belonged to the age group of ≤55 years and mostly they were married (89.7%, n = 269/300). Majorities (59.0%, n = 177/300) of the respondents had graduation or above education and had monthly family income as USD ≤ 853.14 (65.3%, n = 196/300). In addition, nearly half of the study subjects (49%, n = 147/300) were overweight on BMI, while 43.3% were diabetic measured according to the Fasting

Blood Sugar (FBS) level. Surprisingly, the majority had a positive exposure of COVID-19 disease (59.7%, n = 179/300). Furthermore, the study uncovered a considerable prevalence (53%, n = 159/300) of poor lifestyles among the participants which was measured considering the variables like physical activity, sleeping pattern, deleterious habit etc. (Table 2).

### 3.2 Distribution of the components of WHOQOL-BREF in terms of four domains, overall perception of QOL and general health

For further analysis the scores of for domains, perception of overall QOL and general health were categorized in two groups respectively considering the cutoff point 50%. Scores above 50% were computed as 'Good/higher QOL' and below 50% as 'Poor to moderate/lower QOL' [27]. Most of the respondents showed good QOL in both Physical (65%) and environmental (63%) health domain while maximum cases of poor to moderate QOL were found in psychological (71%) and social relationship health domain (74.7%). However, the assessment also showed that most respondents reported a good quality of life in terms of their perception of general health (86.7%) and overall quality of life (77%). (Fig 1).

### 3.3 Spearman's correlation among the components of WHOQOL-BREF with their average distribution

The scores of QOL in the four domains, overall QOL and general health are shown in Table 1. Comparing the four domains of the respondents, environmental health domain was the highest with a mean score of 60.23 ± 8.63 while the social health domain was the lowest with a mean score of 44.90 ± 14.15. Distributions below 50% were considered the cut-off standards for poor QOL. In psychological (DOM 2) and social relationships health (DOM 3) domain majority (71% & 74.7%) showed poor QOL respectively, except environmental (DOM 4) (37%), and physical health domain (35%) which showed good QOL with mean value 60.23 and 53.09 respectively. Spearman's correlation test showed that the four domains, overall perception of QOL (Q1) and overall perception of general health (Q2) were significantly and positively interrelated with predominantly low to moderate relationships (r = 0.13–0.77, $p < 0.01$) (Table 1).

### 3.4 Factors associated with different health domains of QOL

The bivariate cross-tabulation analysis unveiled the basic characteristics of the respondents linked to the four domains of their assessed quality of life. For instance, it was observed that scores DOM1 and DOM2 were significantly (p = 0.01) associated with their age, education, family type, monthly family income, status of DM, history of COVID-19, and lifestyle status of the respondents. In addition, a similar scenario of significant (p = 0.01) associations were observed for DOM3 and DOM4 except education of the subjects. (Table 2).

After that a binary regression analysis was done with the significant variables extracted from the chi-square analysis. Initially, unadjusted odds ratios were revealed against poor QOL of four domains. These are the Crude Odds Ratios (COR) for the poor QOL in four domains respectively. Likewise, higher odds of poor QOL in DOM1 were found among the > 55 years (COR/p = 6.30/0.01) aged diabetic (COR/p = 4.66/0.01) respondents who had a positive history of COVID-19 (COR/p = 4.96/0.01), came from an extended family (COR/p = 3.46/0.01) and had a poor level of lifestyle (COR/p = 30.75/0.01). In addition, significant odds for poor QOL in DOM1 also found among the up to HSC educated subjects (COR/p = 1.71/0.03) who had ≤ 853.14 USD (COR/p = 1.82/0.03) monthly family income. Similarly, robust criteria were revealed for poor QOL in DOM2. Moving towards DOM3 and DOM4, study also revealed some significant factors associated with poor QOL. For instance, significant higher odds for poor QOL in DOM3 and DOM4 were found among the > 55 years aged (poor DOM3: COR/p = 6.72/0.01; DOM4: COR/p = 2.53/0.01) diabetic (poor DOM3: COR/p = 4.05/0.01; DOM4: COR/p = 4.11/0.01) hypertensive patients with poor lifestyle (poor DOM3: COR/p = 20.81/0.01; DOM4: COR/p = 4.11/0.01) who had a positive COVID-19 history (poor DOM3: COR/p = 4.87/0.01; DOM4: COR/p = 2.06/0.01), came from an extended family (poor DOM3: COR/p = 4.29/0.01; DOM4: COR/p = 2.12/0.01) and had ≤ 853.14 USD monthly family income (poor DOM3: COR/p = 3.48/0.01; DOM4: COR/p = 15.67/0.01) (Table 3).

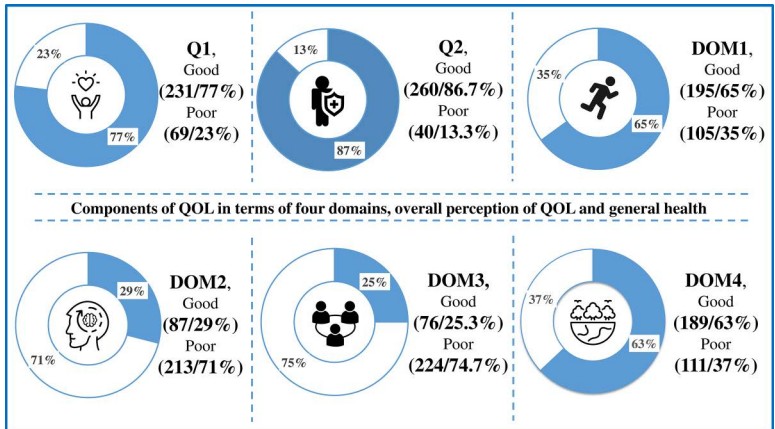

**Fig 1. This is the distribution of different health domains, overall QOL, and general health defining the quality of life among hypertensive respondents (n = 300).** *Data were presented as frequency (n) and percentage (%); Physical health domain (DOM 1); Psychological health domain (DOM 2); Social health domain (DOM 3); Environmental health domain (DOM 4); Overall perception of QOL (Q1); Overall perception of health (Q2); QOL: Quality of life.*

**Table 1. Central distribution with dispersions and correlation coefficients in two overall perceptions and four domains of WHOQOL-BREF (n = 300).**

| Components of QOL | Mean | SD | Number of participants with poor scores[a], n (%) | Q1 | Q2 | DOM 4 | DOM 3 | DOM 2 | DOM 1 |
|---|---|---|---|---|---|---|---|---|---|
| | | | Spearman's correlations (*r*) | | | | | | |
| DOM 1 | 53.09 | 8.36 | 105 (35) | 0.41** | 0.36** | 0.42** | 0.46** | 0.77** | 1 |
| DOM 2 | 47.16 | 11.69 | 213 (71) | 0.32** | 0.31** | 0.62** | 0.65** | 1 | |
| DOM 3 | 44.9 | 14.15 | 224 (74.7) | 0.24** | 0.22** | 0.59** | 1 | | |
| DOM 4 | 60.23 | 8.63 | 111 (37) | 0.13** | 0.25** | 1 | | | |
| Q1 | 57.53 | 12.26 | 69 (23) | 0.42** | 1 | | | | |
| Q2 | 59 | 9.56 | 40 (13.3) | 1 | | | | | |

[a]Scores <1SD, ***p*<0.01, QOL= Quality of Life.

### 3.5 Determinants influencing the different health domains of QOL

After adjusting the model in regression analysis, the study identified key factors influencing poor quality of life across four domains. Model adjustment was performed using a backward elimination approach to control for potential confounders. It was observed that among diabetic patients (AOR/p = 2.35/0.01) aged over 55 (AOR/p = 3.33/0.01) with a history of COVID-19 (AOR/p = 2.97/0.01), poor physical health in DOM1 significantly affected their quality of life. Similar predictors were observed for DOM2 (>55 years age: AOR/p = 2.92/0.01; Income USD ≤ 853.14: AOR/p = 2.34/0.01; diabetic: AOR/p = 2.63/0.01; COVID-19 'yes': AOR/p = 3.14/0.01) and DOM3 (>55 years age: AOR/p = 2.62/0.01; Income USD ≤ 853.14: AOR/p = 2.87/0.01; diabetic: AOR/p = 2.22/0.03; COVID-19 'yes': AOR/p = 3.45/0.01). In DOM4, lower monthly family income (USD ≤ 853.14: AOR/p = 16.89/0.01) emerged as a significant predictor of poor quality of life among elderly (>55 years age: AOR/p = 2.85/0.03) patients. (Table 4).

### 3.6 Factors associated with the perception of overall QOL and general health among the respondents

The study identified older age (COR/p = 2.49/0.01), living in an extended family (COR/p = 2.16/0.01) set up, and having a poor lifestyle (COR/p = 3.23/0.01) as significant factors influencing patients' perception of poor overall quality of life (Q1).

**Table 2. Characteristics of the respondents associated with different health domains of their quality of life (n = 300).**

| Characteristics | Number of participants, n (%) | DOM1 | | | DOM2 | | | DOM3 | | | DOM4 | | |
|---|---|---|---|---|---|---|---|---|---|---|---|---|---|
| | | Good, n (%) | Poor, n (%) | p-value (≤0.05) | Good, n (%) | Poor, n (%) | p-value (≤0.05) | Good, n (%) | Poor, n (%) | p-value (≤0.05) | Good, n (%) | Poor, n (%) | p-value (≤0.05) |
| **Sociodemographic characteristics** | | | | | | | | | | | | | |
| **Age group (in years)** | | | | | | | | | | | | | |
| >55 | 149 (49.7) | 68 (22.7) | 81 (27) | 0.01* | 16 (5.3) | 133 (44.3) | 0.01* | 14 (4.7) | 135 (45) | 0.01* | 78 (26) | 71 (23.7) | 0.01* |
| ≤55 | 151 (50.3) | 127 (42.3) | 24 (8) | | 71 (23.7) | 80 (26.7) | | 62 (20.7) | 89 (29.7) | | 111 (37) | 40 (13.3) | |
| **Education** | | | | | | | | | | | | | |
| Up to H.S.C | 123 (41) | 71 (23.7) | 52 (17.3) | 0.01* | 27 (9) | 96 (32) | 0.01* | 27 (9) | 96 (32) | 0.26 | 76 (25.3) | 47 (15.7) | 0.71 |
| Graduation or above | 177 (59) | 124 (41.3) | 53 (17.7) | | 60 (20) | 117 (39) | | 49 (16.3) | 128 (42.7) | | 113 (37.7) | 64 (21.3) | |
| **Family type** | | | | | | | | | | | | | |
| Nuclear | 170 (56.7) | 131 (43.7) | 39 (13) | 0.01* | 69 (23.3) | 101 (33.7) | 0.01* | 61 (20.3) | 109 (36.3) | 0.01* | 120 (40) | 50 (16.7) | 0.01* |
| Extended | 130 (43.3) | 64 (21.3) | 66 (22) | | 18 (6) | 112 (37.3) | | 15 (5) | 115 (38.5) | | 69 (23) | 61 (20.3) | |
| **Monthly family income (In USD)** | | | | | | | | | | | | | |
| ≤853.14 | 196 (65.3) | 115 (38.3) | 81 (27) | 0.01* | 41 (13.7) | 155 (51.7) | 0.01* | 33 (11) | 163 (54.3) | 0.01* | 92 (30.7) | 104 (34.7) | 0.01* |
| >853.14 | 104 (34.7) | 80 (26.7) | 24 (8) | | 46 (15.3) | 58 (19.3) | | 43 (14.3) | 61 (20.3) | | 97 (32.3) | 7 (2.3) | |
| **Clinical characteristics** | | | | | | | | | | | | | |
| **Status of Diabetes Mellitus (DM)** | | | | | | | | | | | | | |
| Normal | 110 (36.7) | 92 (30.7) | 18 (6) | 0.01* | 53 (17.7) | 57 (19.0) | 0.01* | 45 (15) | 65 (21.7) | 0.01* | 80 (26.7) | 30 (10.0) | 0.01* |
| Pre-diabetic | 60 (20.0) | 35 (11.7) | 25 (8.3) | | 13 (4.3) | 47 (15.7) | | 12 (4) | 48 (16.0) | | 42 (14) | 18 (6) | |
| Diabetic | 130 (43.3) | 68 (22.7) | 62 (20.7) | | 21 (7.0) | 109 (36.3) | | 19 (6.3) | 111 (37.0) | | 67 (22.3) | 63 (21) | |
| **History of COVID-19** | | | | | | | | | | | | | |
| Yes | 179 (59.7) | 93 (31) | 86 (28.7) | 0.01* | 29 (9.7) | 150 (50.0) | 0.01* | 24 (8) | 155 (51.7) | 0.01* | 101 (33.7) | 78 (26) | 0.01* |
| No | 121 (40.3) | 102 (34) | 19 (6.3) | | 58 (19.3) | 63 (21.0) | | 52 (17.3) | 69 (23) | | 88 (29.3) | 33 (11) | |
| **Lifestyle related characteristics** | | | | | | | | | | | | | |
| **Level of lifestyle** | | | | | | | | | | | | | |
| Moderate/healthy | 141 (47) | 134 (44.7) | 7 (2.3) | 0.01* | 80 (26.7) | 61 (20.3) | 0.01* | 69 (23) | 72 (24) | 0.01* | 112 (37.3) | 29 (9.7) | 0.01* |
| Poor | 159 (53) | 61 (20.3) | 98 (32.7) | | 7 (2.3) | 152 (50.7) | | 7 (2.3) | 152 (50.7) | | 77 (25.7) | 82 (27.3) | |

Data are presented as frequency (n), percentage (%); *Statistical significance at p value ≤0.05. The Chi-square test was used to observe the association; QOL.: Quality of life.

**Table 3. Identified predictors associated with four domains of the QOL in unadjusted model (n = 300).**

| Characteristics | DOM1 Poor vs Good | DOM2 Poor vs Good | DOM3 Poor vs Good | DOM4 Poor vs Good |
|---|---|---|---|---|
| | Un-adjusted OR (95% CI)/*p* | Un-adjusted OR (95% CI)/*p* | Un-adjusted OR (95% CI) | Un-adjusted OR (95% CI) |
| **Sociodemographic characteristics** | | | | |
| **Age group (in years)** | | | | |
| > 55 | 6.30 (3.66-10.84)/0.01* | 7.38 (4.01-13.57)/0.01* | 6.72 (3.55-12.73)/0.01* | 2.53 (1.56-4.09)/0.01* |
| ≤ 55 | Reference | Reference | Reference | Reference |
| **Education** | | | | |
| Up to H.S.C | 1.71 (1.06-2.77)/0.03* | 1.82 (1.08-3.09)/0.03* | – | – |
| Graduation or above | Reference | Reference | Reference | Reference |
| **Family type** | | | | |
| Nuclear | Reference | Reference | Reference | Reference |
| Extended | 3.46 (2.11-5.69)/0.01* | 4.25 (2.37-7.63)/0.01* | 4.29 (2.30-7.99)/0.01* | 2.12 (1.32-3.42)/0.01* |
| **Monthly family income (In USD)** | | | | |
| ≤ 853.14 | 2.35 (1.37-4.02)/0.01* | 2.99 (1.79-5.03)/0.01* | 3.48 (2.03-5.97)/0.01* | 15.67 (6.92-35.45)/0.01* |
| > 853.14 | Reference | Reference | Reference | Reference |
| **Clinical characteristics** | | | | |
| **Status of Diabetes Mellitus** | | | | |
| Normal | Reference | Reference | Reference | Reference |
| Pre-diabetic | 3.65 (1.77-7.50)/0.01* | 3.36 (1.64-6.90)/0.01* | 2.77 (1.32-5.79)/0.01* | 1.14 (0.57-2.29)/0.71 |
| Diabetic | 4.66 (2.53-8.50)/0.01* | 4.83 (2.65-8.78)/0.01* | 4.05 (2.18-7.49)/0.01* | 2.51 (1.46-4.31)/0.01* |
| **History of COVID-19** | | | | |
| Yes | 4.96 (2.81-8.78)/0.01* | 4.75 (2.79-8.12)/0.01* | 4.87 (2.78-8.53)/0.01* | 2.06 (1.25-3.38)/0.01* |
| No | Reference | Reference | Reference | Reference |
| **Lifestyle related characteristics** | | | | |
| **Level of lifestyle** | | | | |
| Moderate/healthy | Reference | Reference | Reference | Reference |
| Poor | 30.75 (13.49-70.13)/0.01* | 28.48 (12.45-65.16)/0.01* | 20.81 (9.11-47.56)/0.01* | 4.11 (2.46-6.87)/0.01* |

Logistic Regression Analysis was used to identify the predictors; * Statistical significance at p value ≤0.05, reference category of each domain is good.

While only a poor lifestyle (COR/p = 2.29/0.02) was found to be a significant determinant affecting the perceived general health (Q2) as poor (Table 5).

## 4 Discussion

This study was driven by the need to understand how hypertension—one of the most common chronic conditions in Bangladesh—affects the quality of life of patients in the post-COVID-19 context [6]. While the health risks of hypertension are well recognized, less is known about how it impacts daily life, particularly when combined with socioeconomic challenges and recent experiences with COVID-19 [5,6]. In Bangladesh, where access to healthcare is limited and economic inequalities are widespread, these factors can significantly affect both disease management and overall well-being [4]. This study explored these issues using a multidomain approach to help inform culturally and contextually appropriate strategies that go beyond clinical care. This study supports previous research that has shown that hypertension significantly affects patients' quality of life (QOL) [1,2,28]. Interestingly, our study reported that a higher percentage of participants had good QOL in the physical and environmental health domains [3,29]. This difference may be due to the unique demographic and socioeconomic characteristics of our study population. The high prevalence of COVID-19 exposure among participants is

**Table 4. Identified predictors associated with the four domains of the QOL in adjusted model (n = 300).**

| Characteristics | DOM1 Poor vs Good AOR (95% CI)/p | DOM2 Poor vs Good AOR (95% CI)/p | DOM3 Poor vs Good AOR (95% CI)/p | DOM4 Poor vs Good AOR (95% CI)/p |
|---|---|---|---|---|
| **Age group (in years)** | | | | |
| >55 | 3.33 (1.79-6.17)/0.01* | 2.92 (1.43-5.96)/0.01* | 2.62 (1.24-5.56)/0.01* | Reference |
| ≤55 | Reference | Reference | Reference | 2.85 (1.12-7.26)/0.03* |
| **Monthly family income (In USD)** | | | | |
| ≤853.14 | _ | 2.34 (1.28-4.25)/0.01* | 2.87 (1.55-5.31)/0.01* | 16.89 (7.01-40.73)/0.01* |
| >853.14 | Reference | Reference | Reference | Reference |
| **Status of Diabetes Mellitus** | | | | |
| Normal | Reference | Reference | Reference | Reference |
| Pre-diabetic | 1.78 (0.79-3.96)/0.16 | 1.66 (0.73-3.76)/0.23 | 1.34 (0.57-3.13)/0.50 | – |
| Diabetic | 2.35 (1.17-4.71)/0.02* | 2.63 (1.32-5.25)/0.01* | 2.22 (1.08-4.55)/0.03* | – |
| **History of COVID-19** | | | | |
| Yes | 2.97 (1.59-5.56)/0.01* | 3.14 (1.70-5.78)/0.01* | 3.45 (1.82-6.53)/0.01* | – |
| No | Reference | Reference | Reference | Reference |

Logistic Regression Analysis was used to identify the predictors; * Statistical significance at p value ≤0.05, AOR=Adjusted Odds Ratio, reference category of each domain is good.

**Table 5. Characteristics of the respondents associated with their perception of overall QOL and general health (n = 300).**

| Characteristics | Overall general QOL (Q1) Poor Vs Good | | | | General health QOL (Q2) Poor Vs Good | | | |
|---|---|---|---|---|---|---|---|---|
| | Good, n (%) | Poor, n (%) | p-value (≤0.05) | COR (95% CI)/p | Good, n (%) | Poor, n (%) | p-value (≤0.05) | COR (95% CI)/p |
| **Age group (in years)** | | | | | | | | |
| >55 | 103 (34.3) | 46 (15.3) | 0.01* | 2.49 (1.41-4.37)/0.01* | 124 (41.3) | 25 (8.3) | 0.08 | 1.83 (0.92-3.63)/0.08 |
| ≤55 | 128 (42.7) | 23 (7.7) | | Reference | 136 (45.3) | 15 (5) | | Reference |
| **Family type** | | | | | | | | |
| Nuclear | 141 (47) | 29 (9.7) | 0.01* | Reference | 69 (23.3) | 101 (33.7) | 0.11 | Reference |
| Extended | 90 (30) | 40 (13.3) | | 2.16 (1.25-3.73)/0.01* | 18 (6) | 112 (37.3) | | 1.72 (0.88-3.36)/0.11 |
| **Level of lifestyle** | | | | | | | | |
| Moderate/healthy | 123 (41) | 18 (6) | 0.01* | Reference | 80 (26.7) | 61 (20.3) | 0.02 | Reference |
| Poor | 108 (36) | 51 (17) | | 3.23 (1.78-5.86)/0.01* | 7 (2.3) | 152 (50.7) | | 2.29 (1.12-4.71)/0.02* |

Data are presented as frequency (n), percentage (%); *Statistical significance at p value ≤0.05. Chi-square test was used to observe the association; QOL: Quality of life, Logistic Regression Analysis was used to identify the predictors; reference category for overall general QOL and general health was 'Good'.

a unique aspect of our study [5,30], which has not been extensively explored in previous studies [6,31]. The influence of COVID-19 on QOL requires further investigation for the effective management of hypertensive patients [7,32]. Additionally, our study highlighted a significant prevalence of poor lifestyles among participants [8,33], suggesting an urgent need for tailored interventions to improve lifestyle behavior in hypertensive patients. This finding is consistent with the prior RICH LIFE project, which emphasized the importance of addressing social and medical needs in managing hypertension [9,34].

Our findings indicated a significant correlation between various domains of the WHOQOL-BREF, overall QOL, and general health, which align with prior research [10,35]. However, our study revealed a notable finding for hypertensive

patients by demonstrating the impact of COVID-19 history on QOL, adding new insights to existing QOL research [11,36]. A study conducted in Ethiopia found better QOL in female hypertensive patients [12], whereas our study did not recognize gender as a significant factor. This discrepancy may be due to cultural or regional differences that influence perceptions of QOL. The current study found a significant association between age, education, family type, and lifestyle status with QOL domains, which aligns with previous research [13,14,28]. However, our study found a strong association between poor QOL and lower monthly family income (≤853.14 USD), highlighting the significant role of socioeconomic status in health outcomes [15,29,31].

This study adds valuable insights to the existing research on the quality of life in hypertensive patients, particularly in the context of the COVID-19 pandemic. It highlights the need for holistic patient care that considers various demographic and lifestyle factors [16,32,33]. By incorporating patients' COVID-19 history, this research provides a deeper understanding of the quality of life in hypertensive patients. Additionally, focusing on socioeconomic factors such as income levels emphasizes the need for targeted interventions to improve quality of life among these populations [17,30]. The current study's findings supported the Bushehr Elderly Health Program [18], which found that diabetes significantly impacts QOL in older individuals. However, our study uniquely considers the history of COVID-19 in addressing hypertensive patients, a factor not explored in the Bushehr study [19].

Conversely, a systematic review of QOL in type 2 diabetic patients [20] revealed that non-medical factors (physical activity and socio-demographic factors) are the most important determinants of QOL. This result contrasts with our study, which indicates that medical status and income are key factors of QOL [21]. In addition, research on QOL for diabetic patients in the context of COVID-19 [22] supports our results, especially in identifying the impact of COVID-19 on QOL. Unlike our study, it did not explore differences in QOL based on income levels. Our study provides valuable evidence considering the different factors influencing patients' QOL, particularly the holistic aspect of COVID-19 and different socio-economic backgrounds [23,24]. These findings extend the current literature, suggesting the importance of holistic patient care that considers both medical and socioeconomic aspects [25,27].

Bangladesh has a high prevalence of hypertension and significant public health challenges exacerbated by economic disparities. The high prevalence of COVID-19 affected the quality of life for hypertensive patients, emphasizing the urgent need to integrate COVID-19 management with hypertension care. Insights from this study can guide health promotion strategies tailored to the needs of these patients. Besides, it reinforces the importance of combining COVID-19 care with hypertension treatment and addressing the socioeconomic factors that affect health outcomes.

The study included a diverse group of participants from different genders, education levels, and income brackets that enhanced the applicability of the results. Utilizing the WHOQOL-BREF instrument provided a detailed evaluation across multiple domains of quality-of-life, such as physical health, mental well-being, environmental health, and social relationships. In the Bangladeshi context, prior studies have supported the construct validity of the WHOQOL-BREF using more robust statistical methods. For example, Amin et al. (2022) validated the tool among type 2 diabetes patients using confirmatory factor analysis (CFA), while Islam (2025) applied both CFA and item-level analyses to confirm the instrument's multidimensional structure among older adults [10,11]. These findings reinforce the psychometric soundness of the WHOQOL-BREF for use in diverse Bangladeshi populations. We recommend that future research employing Rasch analysis or more comprehensive CFA models in hypertensive populations would further strengthen the evidence base for its use in this context.

Several limitations should be acknowledged. First, cross-sectional design restricts our ability to infer causality. Second, as the study was conducted in a private tertiary hospital in urban Dhaka, the findings may not be fully generalizable to hypertensive populations in rural or underserved areas. The reliance on self-reported data also introduces the potential for reporting bias. Additionally, due to the absence of local data on binary QOL outcomes, we approximated the sample size using an average WHOQOL-BREF domain score (0.76) drawn from published literature. While this approach was practical, it remains an approximation. Future studies would benefit from pilot data or the use of stratified QOL thresholds to support more precise, context-specific sample size calculations.

Despite these limitations, the study has several strengths. It was conducted in a well-resourced tertiary care setting that offers structured follow-up and equitable access to services for both men and women. This may have mitigated gender-related disparities in care and health outcomes often seen in less-structured, community-based settings. Although urban healthcare facilities can limit generalizability, the consistency of care and documentation in such settings reduced variability and enhanced the reliability of our findings. As private hospitals become increasingly integrated into Bangladesh's national NCD care framework, understanding patient outcomes in these settings is both timely and policy relevant. Rather than constraining generalizability, the study highlights key determinants of QOL—such as age, diabetes, income, and COVID-19 history—that are likely to be relevant across diverse healthcare contexts.

We also employed a validated multidimensional tool (WHOQOL-BREF), used systematic random sampling, and included participants from varied sociodemographic backgrounds, all of which enhanced the internal validity of the study. Furthermore, the use of Spearman's correlation and regression analyses allowed for a robust examination of factors associated with QOL. Importantly, this study adds new insights to the existing literature by exploring the impact of COVID-19 exposure on the quality of life of hypertensive patients. Here, the "post-pandemic" period refers to March–June 2023—a time when Bangladesh had exited the acute phase of the pandemic, but many individuals, especially those with chronic conditions, continued to face challenges such as delayed care, financial strain, and psychological stress. This period represents a distinct context from both the peak-pandemic and pre-pandemic phases and offers meaningful insights into how lingering challenges impact the quality of life in hypertensive patients. By examining this context, the study sheds light on the lived experiences of a population whose quality of life—impacted by both chronic illness and pandemic-related stressors—has been largely underexplored in current literature.

## 5 Conclusion

Study revealed a significant poor QOL in the psychological and social domains among the hypertensive patients. Poor QOL across all domains was significantly associated with older age, diabetes, prior COVID-19 infection, and lower monthly family income. These factors reflect not only clinical vulnerability but also cultural and socioeconomic conditions in Bangladesh, such as limited access to care, dietary constraints, and caregiving roles within multigenerational families. Moreover, older age and poor lifestyle were also significantly associated with poor overall quality of life and general health perception of the respondents. These findings appeal the need for culturally grounded strategies, including family-based health education, peer support for older adults, and subsidized services for low-income patients, to enhance patient-centered care beyond clinical treatment. To enhance the quality of life for hypertensive patients, implementation of effective interventions considering the determinants are crucial. Strengthening of health care service considering QOL in the cardiac departments may reduce the hypertension-related burdens and contribute to achieving Sustainable Development Goal 3 by minimizing the impact of non-communicable diseases.

## Supporting information

**S1 Checklist. STROBE_checklist_cross-sectional.**
(DOCX)

**S1 File. Ethical Letter NUB Nasrin QOL.**
(PDF)

**S2 File. PLOSOne_Human_Subjects_Research_Checklist (1).**
(DOCX)

**S1 Data. Final Data set_for article.sav=.**
(XLS)

## Author contributions

**Conceptualization:** Nasrin Akter, Bilkis Banu, Fatema Afrin Kanta, Sarder Mahmud Hossain.

**Data curation:** Nasrin Akter, Farhana Faruque Zerin, Bilkis Banu, Fatema Afrin Kanta, Shahnaz Begam, Sarder Mahmud Hossain.

**Formal analysis:** Nasrin Akter.

**Funding acquisition:** Nasrin Akter, Farhana Faruque Zerin, Fatema Afrin Kanta.

**Investigation:** Nasrin Akter, Farhana Faruque Zerin, Bilkis Banu, Shahnaz Begam, Sarder Mahmud Hossain.

**Methodology:** Nasrin Akter, Farhana Faruque Zerin, Bilkis Banu, Fatema Afrin Kanta.

**Project administration:** Sarder Mahmud Hossain.

**Resources:** Nasrin Akter, Farhana Faruque Zerin, Fatema Afrin Kanta, Shahnaz Begam.

**Software:** Nasrin Akter.

**Supervision:** Nasrin Akter, Farhana Faruque Zerin, Bilkis Banu, Sarder Mahmud Hossain.

**Validation:** Nasrin Akter, Farhana Faruque Zerin, Bilkis Banu, Fatema Afrin Kanta, Shahnaz Begam, Sarder Mahmud Hossain.

**Visualization:** Farhana Faruque Zerin, Bilkis Banu, Fatema Afrin Kanta, Shahnaz Begam, Sarder Mahmud Hossain.

**Writing – original draft:** Nasrin Akter, Farhana Faruque Zerin, Bilkis Banu, Fatema Afrin Kanta, Shahnaz Begam, Sarder Mahmud Hossain.

**Writing – review & editing:** Nasrin Akter, Farhana Faruque Zerin, Bilkis Banu, Fatema Afrin Kanta, Shahnaz Begam, Sarder Mahmud Hossain.

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
