## [Decision Letter · Decision Letter 0]

20 May 2025

Dear Dr. Akter,

Thank you for submitting your manuscript to PLOS ONE. After careful consideration, we feel that it has merit but does not fully meet PLOS ONE’s publication criteria as it currently stands. Therefore, we invite you to submit a revised version of the manuscript that addresses the points raised during the review process.

We look forward to receiving your revised manuscript.

Kind regards,

Md. Feroz Kabir, BPT, MPT, MPH, BPED, MPED

Academic Editor

PLOS ONE

2. We note that your Data Availability Statement is currently as follows: [All relevant data are within the manuscript and its Supporting Information files]

Additional Editor Comments:

Please submit the revised manuscript within the next 15 days. Please carefully revise the manuscript according to the comments of the reviewers.

Reviewers' comments:

Reviewer's Responses to Questions

**Comments to the Author**

1. Is the manuscript technically sound, and do the data support the conclusions?

Reviewer #1: Yes

Reviewer #2: Yes

Reviewer #3: Yes

Reviewer #4: No

2. Has the statistical analysis been performed appropriately and rigorously?

Reviewer #1: No

Reviewer #2: No

Reviewer #3: Yes

Reviewer #4: No

3. Have the authors made all data underlying the findings in their manuscript fully available?

Reviewer #1: Yes

Reviewer #2: Yes

Reviewer #3: Yes

Reviewer #4: Yes

4. Is the manuscript presented in an intelligible fashion and written in standard English?

Reviewer #1: Yes

Reviewer #2: Yes

Reviewer #3: Yes

Reviewer #4: Yes

Reviewer #1: The manuscript focuses on a health issue that is highly relevant for research and practical purposes.

1. Although the manuscript is clear and for the most part coherent, the novelty aspect is quite weak. Thus, I find it hard to recommend it for publication in its current form. It seems that the gap is related to the the lack of research to "provide culturally relevant insights to guide effective healthcare interventions and improve the well-being of individuals living with hypertension.". The authors did not make it clear how the existing literature is lacking in this matter. Additionally, how does measuring QOL would fill this gap? What other alternatives are there to solve this problem? Is compiling QOL data the most effective way to solve the problems? I highly doubt the assertion that "Without this data, it’s hard to develop effective and culturally appropriate strategies to support and improve the well being of hypertensive patients". There are already a strong body of knowledge to be used. If it is indeed culturally inappropriate for Bangladeshi context, then the authors have to explain so.

A stronger and clearer argument need to be written to highlight the novelty of this study.

2. From a research literature perspective, it is not clear why hypertensive patients were chosen as the focus of the study. How are the patients different enough compared to other patients? Why can't the authors simply use existing studies on QOL of patients with various illnesses to "develop effective and culturally appropriate strategies".

3. Conclusion section: "As predictor, poor QOL in all domains were found significant among the older, diabetic patients who had history of COVID-19, and poor monthly family income." I have 2 concerns about this conclusion:

a. If the authors want to address cultural nuances, how does these significant predictors relate to cultural uniqueness of Bangladesh? In other words, what culture specific findings are there?

b. How will these findings help in designing effective strategies? Increase the patients' income? Make them younger? Cure their diabetes? I can't see how the identification of factors influencing QOL can help in designing such strategies.

4. The method needs more details (e.g. what kind of random selection method was used? how long had the patients have hypertension? What are the levels of hypertension experienced by the patients? What language was used for the WHOQOL?)

5. What previous validity evidence are there regarding the use of WHOQOL-BREF among Bangladeshi? Internal consistency index (Cronbach alpha) is not a strong evidence for the validity of the scores. Analyses like CFA or Rasch Rating Scale Model are more robust to convince readers of the validity of the scores.

6. For the regression analysis, assumption checks (and the results) need to be stated.

7. The Discussion need to be revamped following a clearer research problem statement (not necessarily 'research gap').

8. The focus on patients with hypertension seem to be diluted given the findings on COVID-19 and diabetes status of the patients. Any conclusion drawn from the study cannot be stated as representing patients with hypertension. The manuscript title is therefore not reflecting the findings.

While the manuscript does not show any fatal flaw, I find it lacking novelty that warrant a publication in a journal. It works well as a routine contemporary updates for Bangladeshi market.

Reviewer #2: The manuscript reads well. Please consider the following minor comments when preparing the revised manuscript.

Typos

"...DOM1 and DOM1 were significantly (p=0.01) associated with their age, education, family type, monthly..."

Methods

1. "...the study was conducted at a randomly selected hospital (Square Hospitals

Limited) from the top 10 hospitals..." Could you please identify the other nine private hospitals included in the study? What criteria were used to rank these hospitals, and what was the rationale for this ranking methodology? Furthermore, what were the reasons for focusing on private rather than public hospitals in this research? Given that private healthcare in Bangladesh can be expensive and often requires insurance, how does the study account for potential biases in representing higher-income populations?

2. how authors come up with USD853.14 as good or bad household income.

3. "...A random sampling technique was employed in this study to ensure the generalizability of the findings." The total number of eligible patients from which the random sample of 300 was drawn. So. the initial patient pool is missing from that statement alone.

4. "...The overall prevalence of hypertensive patients in the Dhaka reflected 31%..."

The observed hypertension prevalence of 31% in Dhaka suggests that odds ratios from logistic regression may not be a good approximation of prevalence ratios. Could you elaborate on why logistic regression was chosen over methods known to directly estimate prevalence ratios, such as Poisson regression with robust standard errors?

5. Include strength and weakness of the study in discussion section

Reviewer #3: The manuscript under review offers a reasonably thorough examination of public health patterns, supported by an adequate sample size (n=300), which meets the conventional minimum threshold for statistical significance. The authors' application of statistical methods throughout the paper is generally appropriate (lines 24–28), though there are persistent issues with terminology usage—most notably the incorrect treatment of plural forms such as "data" (line 22) and "statistics" (lines 24, 28), which detract from the academic rigor.

Early in the manuscript, clarity in definitions would benefit the general readership. For instance, the description in line 43 should be revised to read: “Hypertension, commonly referred to as high blood pressure,” as “hypertension” is the correct clinical term.

There are more substantive concerns regarding methodological transparency. On line 119, the claim that Dhaka was selected through “multi-stage random sampling” raises doubts, particularly when the authors themselves cite the city’s extensive healthcare infrastructure as a benefit. This contradiction suggests convenience sampling may have played a role, undermining the generalizability of the findings.

Similarly, the justification provided for the sample size (line 126) lacks credibility. While the authors imply that their choice of 300 participants is a deliberate methodological decision, it more likely reflects the standard threshold for statistical relevance. Further clarification is needed to distinguish between statistical necessity and genuine methodological reasoning.

The paper mentions that the data collection instrument was “pre-tested” (line 133), but fails to specify critical details—namely, the demographic or size of the test group. It is unclear whether the pre-test was conducted among the study’s target population or a separate group, which raises questions about the reliability and validity of the questionnaire.

On a conceptual level, the discussion around COVID-19 exposure (lines 195–196) appears disjointed. The finding that 59.7% of participants had been exposed to COVID-19 is not inherently surprising and lacks a compelling rationale or comparison benchmark that would make this statistic meaningful.

In terms of structure, there is some redundancy toward the end of the manuscript (lines 321–327), where information is repeated unnecessarily. Additionally, the paper concludes with a statement of limitations (line 329), which, while important, may not be the most impactful way to close the manuscript. A more compelling conclusion could emphasize the broader implications or recommendations, while still incorporating the study’s limitations.

A final critical consideration is whether the paper offers culturally appropriate strategies, as alluded to earlier in the text (line 85). This is a potentially valuable contribution, yet it remains unclear whether such strategies are proposed or substantiated in the discussion.

In summary, the manuscript provides a solid foundation in terms of data collection and statistical analysis but is weakened by inconsistencies in methodological transparency, terminological imprecision, and rhetorical structure. Overall, the study offers a meaningful contribution to multiple disciplines and thus has inherent value to the research community writ large.

Reviewer #4: This manuscript investigates the quality of life (QOL) among hypertensive patients in a selected tertiary hospital in Dhaka, Bangladesh, using the WHOQOL-BREF instrument.

While the study addresses an interesting public health question, it suffers from multiple fundamental flaws related to the representativeness of the study population. Due to these issues in the selection of a highly specific and unrepresentative study site (Square Hospitals Limited), which severely compromises the generalizability and external validity of the findings for the broader hypertensive population in Dhaka, or Bangladesh, this manuscript is not suitable for publication. The conclusions drawn cannot reliably inform public health strategies for the majority of the population. Combined with other methodological and interpretative concerns, these issues warrant Rejection.

A. Fundamental Issues:

1. The study was conducted at Square Hospitals Limited, described as a "leading tertiary care facility" and one of the "top 10 hospitals" in Dhaka with a dedicated cardiac department. Such premier institutions in Dhaka are only accessible primarily to a more affluent segment of the population. This deliberate selection of a high-end, private hospital introduces a significant selection bias. The patient population from such a facility is unlikely to be representative of the general hypertensive population in Dhaka, and certainly not of Bangladesh, where a substantial portion of the populace is underprivileged and relies on public healthcare facilities or less expensive private options.

2. Consequently, the findings regarding QOL and its determinants from this specific, potentially socioeconomically advantaged, patient cohort cannot be generalized to the broader population of hypertensive patients in Bangladesh. The study's conclusions and recommendations, if based on this unrepresentative sample, may be misleading for public health planning and interventions aimed at the general population. The manuscript aims to provide insights for "Bangladesh" (Abstract, Background, Conclusion), but the sampling strategy fundamentally undermines this goal.

3. The manuscript does not adequately acknowledge or discuss the profound limitations imposed by this choice of study site on the external validity of its findings. While the discussion mentions "focusing on specific region (Dhaka) may limit the generalizability," this does not capture the critical issue of socioeconomic selectivity within Dhaka due to the choice of hospital. Furthermore, the authors’ interpretation in multiple instances appear quite out of scope.

B. Rationale and Contextualization

1. While the study mentions the COVID-19 pandemic's impact, the "post-pandemic" aspect could be more clearly defined and integrated. It's unclear what specific timeframe "post-pandemic" refers to in this study and how this phase distinctly influences QOL compared to the pandemic phase or pre-pandemic times.

2. The rationale for choosing WHOQOL-BREF over other instruments like EQ-5D (which the authors mention was used in Indonesia during the pandemic) could be strengthened, particularly in the context of previous HRQOL studies in Bangladesh.

C. Methods

1. Patients were "randomly selected from two departments... using the patient register." Further details on the randomization process are needed. Were all patients in the register eligible? How was systematic bias avoided? Are these two departments (Cardiac & Vascular Surgery and Cardiology) representatives of all hypertensive patients, even within this hospital (e.g., what about patients managed in general medicine or outpatient clinics for hypertension without acute cardiac events)?

2. The proportion (p) used for sample size calculation was 0.76, stated as "the QOL among hypertensive patients expressed by mean score of domains ranging from 0.65 to 0.88, with 0.76 considered as the average" (citing a study from rural Vietnam). Using an average of a range of mean scores as a 'proportion' for sample size calculation for a binary outcome (implied by later categorizing QOL as 'Good/higher' vs 'Poor to moderate/lower') is methodologically questionable. The justification for p=0.76 needs to be clearer and more appropriate for the intended analysis.

3. The manuscript states that poor lifestyle was "measured considering the variables like physical activity, sleeping pattern, deleterious habit etc." (Results, Table 2 footnote). This is vague. The specific components, how they were measured, and how they were combined to classify "poor lifestyle" need to be explicitly detailed in the methods.

4. The scores for QOL domains were dichotomized using 50% as a cutoff ("Scores above 50% were considered as 'Good/higher QOL'; while below 50% as 'Poor to moderate/lower QOL'"). This transformation of a continuous scale into a binary variable can lead to loss of information. Justification for this specific cutoff and the potential impact of this dichotomization should be discussed.

5. Statistical Analysis: The use of backward elimination for regression analysis can be problematic as it may lead to model instability and exclusion of important confounders based on statistical significance rather than theoretical importance.

D. Interpretation and Discussion

1. Given the significant limitation of the study site, any conclusions drawn about the QOL of hypertensive patients in Bangladesh are likely overstated. The discussion should be heavily tempered by this limitation.

2. When comparing findings (e.g., gender not being significant vs. an Ethiopian study where it was), the discussion should consider that the highly selective nature of the current study's sample might explain these discrepancies, rather than just "cultural or regional differences."

3. The study found lower monthly family income was associated with poor QOL. In a sample drawn from a high-end hospital, this "lower income" group is still relatively well-off compared to the general population, making the interpretation of this finding misleading.

**Do you want your identity to be public for this peer review?** For information about this choice, including consent withdrawal, please see our Privacy Policy

Reviewer #1: No

Reviewer #2: **Yes:** Proloy Barua

Reviewer #3: No

Reviewer #4: No

---

## [Author Response · Author response to Decision Letter 1]

3 Jul 2025

Thanks a lot for the valuable feedback with the revision requests. We tried our best to update the manuscript accordingly.

Reviewer #1: The manuscript focuses on a health issue that is highly relevant for research and practical purposes.

1. Although the manuscript is clear and for the most part coherent, the novelty aspect is quite weak. Thus, I find it hard to recommend it for publication in its current form. It seems that the gap is related to the lack of research to "provide culturally relevant insights to guide effective healthcare interventions and improve the well-being of individuals living with hypertension." The authors did not make it clear how the existing literature is lacking in this matter. Additionally, how does measuring QOL would fill this gap? What other alternatives are there to solve this problem? Is compiling QOL data the most effective way to solve the problems? I highly doubt the assertion that "Without this data, it’s hard to develop effective and culturally appropriate strategies to support and improve the well-being of hypertensive patients". There are already a strong body of knowledge to be used. If it is indeed culturally inappropriate for Bangladeshi context, then the authors have to explain so.

A stronger and clearer argument need to be written to highlight the novelty of this study.

Thank you for your feedback. To address this, we have revised the last two paragraphs of the Background section to more clearly articulate the novelty of our study. We clarified that while there is existing global knowledge on QOL in chronic conditions, studies specifically using WHOQOL-BREF in hypertensive populations in post-pandemic Bangladesh remain limited. We also justified why QOL data is critical for designing culturally relevant strategies—something not possible with generalized data from other diseases or global contexts (Line 81-85). As noted in the discussion section, during the pandemic, the European Quality of Life-5 Dimensions (EQ-5D) tool was used to assess quality of life among hypertensive patients in Indonesia. To address your query about alternatives, we added a comparison of other methods such as EQ-5D and HRQoL tools and justified why WHOQOL-BREF was selected. To further contextualize our choice of instrument, the following paragraph is added (Line no 76-84)-

“While tools such as the EQ-5D are widely used for assessing QOL, they primarily focus on health utility values and are more suitable for economic evaluations. In contrast, WHOQOL-BREF offers a broader, multidimensional assessment—including physical, psychological, social, and environmental domains, making it more appropriate for understanding the lived experiences of hypertensive patients in a culturally nuanced context. Additionally, the WHOQOL-BREF has been cross-culturally validated and shown to be adaptable for use in LMICs like Bangladesh, supporting its relevance for this study”.

2. From a research literature perspective, it is not clear why hypertensive patients were chosen as the focus of the study. How are the patients different enough compared to other patients? Why can't the authors simply use existing studies on QOL of patients with various illnesses to "develop effective and culturally appropriate strategies".

Thank you for your insightful observation. We have now explicitly clarified the rationale for selecting hypertensive patients as the focus of our study. Although QOL research exists for patients with various chronic illnesses, hypertension presents distinct clinical and psychosocial challenges. As a chronic asymptomatic condition with serious long-term consequences, hypertension often lacks adequate patient engagement, leading to reduced treatment adherence and quality of life. Furthermore, the psychological and social impacts of hypertension, particularly in post-pandemic Bangladesh, remain under-researched, especially using multidimensional tools like WHOQOL-BREF. Our revised manuscript now explains why general QOL studies cannot be directly applied to hypertensive populations without cultural and contextual adaptation. This justification is now reflected clearly in the background section. The following paragraph is added in the background section (Line no 86-96)-

“While QOL research among chronic illnesses exists, hypertension warrants distinct attention due to its asymptomatic nature and long-term behavioral management demands. Unlike conditions that present with immediate symptoms (e.g., diabetes or stroke), hypertension’s “silent” progression can result in patients underestimating its seriousness, thereby diminishing adherence to treatment and engagement with health services. Moreover, the post-pandemic psychosocial stressors—such as social isolation, fear of comorbid risk, and disrupted routines—may uniquely impact individuals with hypertension. Thus, generalized QOL findings from other illness groups may not capture the nuanced experiences of hypertensive patients in Bangladesh. This study specifically addresses this gap by exploring their QOL through a multidimensional and culturally adaptable tool”.

3. Conclusion section: "As predictor, poor QOL in all domains were found significant among the older, diabetic patients who had history of COVID-19, and poor monthly family income." I have 2 concerns about this conclusion:

a. If the authors want to address cultural nuances, how does these significant predictors relate to cultural uniqueness of Bangladesh? In other words, what culture specific findings are there?

b. How will these findings help in designing effective strategies? Increase the patients' income? Make them younger? Cure their diabetes? I can't see how the identification of factors influencing QOL can help in designing such strategies.

Thank you for raising these thoughtful concerns. We agree that the cultural relevance of our findings and their practical implications should be more explicitly stated. The predictors—older age, diabetes, COVID-19 history, and lower income—are not just biological or economic markers; they intersect with social norms, access to care, and health-seeking behaviors in Bangladesh. For example, older adults in extended families may prioritize others’ health over their own, and stigma around chronic disease or mental health can deter engagement with care. Limited financial resources often restrict dietary options and continuity of treatment. Thus, these predictors reflect cultural and structural vulnerabilities that influence health outcomes, rather than fixed individual attributes.

Rather than “making patients younger” or “curing diabetes,” our intent is to guide culturally tailored strategies—like age-appropriate psychosocial support, community-based diabetes education, or income-sensitive intervention packages (e.g., subsidized nutrition counseling or mobile health services). These findings help stakeholders identify high-risk subgroups for targeted resource allocation, and they emphasize the need to go beyond pharmacologic care by addressing social determinants within culturally meaningful frameworks.

The following paragraph is added (Line 455-458):

Poor QOL across all domains was significantly associated with older age, diabetes, prior COVID-19 infection, and lower monthly family income. These factors reflect not only clinical vulnerability but also cultural and socioeconomic conditions in Bangladesh, such as limited access to care, dietary constraints, and caregiving roles within multigenerational families.

Again, the following paragraph is added (Line 460-463):

These findings appeal the need for culturally grounded strategies, including family-based health education, peer support for older adults, and subsidized services for low-income patients, to enhance patient-centered care beyond clinical treatment.

4. The method needs more details (e.g. what kind of random selection method was used? how long had the patients have hypertension? What are the levels of hypertension experienced by the patients? What language was used for the WHOQOL?)

We appreciate your valuable feedback. In response, we have revised the Methods section to clarify the random sampling technique, participants’ duration of hypertension, classification of hypertension levels, and the language used in administering the WHOQOL-BREF instrument.

Specific Revisions Made:

1. For addressing random Selection Method concern the following below is added in section 2.2 (Line 152-155)

“Initially, 600 hypertensive inpatients were identified from the admission registry of the Cardiac & Vascular Surgery and Cardiology (Interventional) departments. From this pool, every second patient was selected using systematic random sampling, resulting in a final sample of 300 participants.”

2. For addressing duration of Hypertension concern the following below is added in section 2.2 (Line 133-135)

“Participants had been diagnosed with hypertension for a minimum duration of six months prior to the survey, based on hospital records.”

3. For addressing levels of Hypertension concern the following below is added in section 2.3 (Line 184-186)

“Hypertension levels were categorized using JNC 8 criteria into Stage 1 (systolic 140–159 mmHg or diastolic 90–99 mmHg) and Stage 2 (systolic ≥160 mmHg or diastolic ≥100 mmHg) based on recent clinical measurements retrieved from patient charts.”

4. For language of WHOQOL-BREF – The following below is added in section 2.3 (Line 179-181)

“The WHOQOL-BREF was administered in Bengali, the native language of participants, ensuring cultural and linguistic appropriateness.”

5. What previous validity evidence are there regarding the use of WHOQOL-BREF among Bangladeshi? Internal consistency index (Cronbach alpha) is not a strong evidence for the validity of the scores. Analyses like CFA or Rasch Rating Scale Model are more robust to convince readers of the validity of the scores.

We sincerely thank you for the valuable comment. While our study reported satisfactory internal consistency using Cronbach’s alpha across the WHOQOL-BREF domains, we acknowledge that this metric is not strong evidence for the validity of the scores. In response, we have revised the following paragraph (line no 412- 420)-

“In the Bangladeshi context, prior studies have supported the construct validity of the WHOQOL-BREF using more robust statistical methods. For example, Amin et al. (2022) validated the tool among type 2 diabetes patients using confirmatory factor analysis (CFA), while Islam (2025) applied both CFA and item-level analyses to confirm the instrument’s multidimensional structure among older adults. These findings reinforce the psychometric soundness of the WHOQOL-BREF for use in diverse Bangladeshi populations. However, we agree that future research employing Rasch analysis or more comprehensive CFA models in hypertensive populations would further strengthen the evidence base for its use in this context”.

References:

Amin MF, Bhowmik B, Rouf R, Khan MI, Tasnim SA, Afsana F, Sharmin R, Hossain KN, Khan MA, Amin SM, Khan MS. Assessment of quality of life and its determinants in type-2 diabetes patients using the WHOQOL-BREF instrument in Bangladesh. BMC endocrine disorders. 2022 Jun 18;22(1):162.

Islam FM. Gender difference in domain-specific quality of life measured by modified WHOQoL-BREF questionnaire and their associated factors among older adults in a rural district in Bangladesh. PloS one. 2025 Jan 7;20(1):e0317113.

6. For the regression analysis, assumption checks (and the results) need to be stated.

Thank you for your valuable suggestion. We have now updated the Data Analysis section (Section 2.5) of the manuscript to include information on the assumption checks conducted prior to regression analysis. Specifically, we assessed multicollinearity, independence of residuals, and model fit to ensure the validity of the logistic regression results. The methods and outcomes of these checks have been added to ensure transparency and analytical rigor. The following paragraph are added (Line 253-260)

Before performing binary logistic regression, relevant assumptions were checked. Multicollinearity was evaluated using Variance Inflation Factor (VIF), and all predictors showed VIF values below 2.0, indicating no multicollinearity. Independence of residuals was examined using the Durbin-Watson statistic, which fell within the acceptable range of 1.6 to 2.2. Additionally, the goodness-of-fit of the regression models was evaluated using the Hosmer–Lemeshow test, with p-values greater than 0.05 suggesting a good model fit. These checks confirmed the suitability of the data for regression analysis.

7. The Discussion need to be revamped following a clearer research problem statement (not necessarily 'research gap').

Thank you for your thoughtful feedback. To address this, we revised the beginning of the Discussion section to include a clear restatement of the research problem that guided our study. We clarified that the study aimed to examine how hypertension affects the quality of life in the sociocultural and post-pandemic context of Bangladesh—particularly given the country’s unique challenges related to healthcare access, economic inequality, and COVID-19 exposure. This framing better situates the findings within the broader public health context and highlights the study’s purpose beyond simply filling a literature gap. The following paragraph are added (Line 352-360)-

“This study was driven by the need to understand how hypertension—one of the most common chronic conditions in Bangladesh—affects the quality of life of patients in the post-COVID-19 context. While the health risks of hypertension are well recognized, less is known about how it impacts daily life, particularly when combined with socioeconomic challenges and recent experiences with COVID-19. In Bangladesh, where access to healthcare is limited and economic inequalities are widespread, these factors can significantly affect both disease management and overall well-being. This study explored these issues using a multidomain approach to help inform culturally and contextually appropriate strategies that go beyond clinical care”.

8. The focus on patients with hypertension seem to be diluted given the findings on COVID-19 and diabetes status of the patients. Any conclusion drawn from the study cannot be stated as representing patients with hypertension. The manuscript title is therefore not reflecting the findings. While the manuscript does not show any fatal flaw, I find it lacking novelty that warrant a publication in a journal. It works well as a routine contemporary updates for Bangladeshi market.

We sincerely thank you for the thoughtful feedback. However, we respectfully disagree with the assertion that the focus on hypertensive patients has been diluted. The manuscript specifically recruited participants with a confirmed diagnosis of hypertension, and the primary objective was to assess their quality of life (QOL) using the WHOQOL-BREF instrument. Comorbidities such as diabetes and history of COVID-19 were analyzed not as focal populations but as covariates, reflecting real-world complexities in hypertensive care. These factors were included to examine their modifying effects on QOL outcomes among individuals with hypertension—not to shift the study's central focus.

This analytic strategy enhances the ecological validity of the findings and acknowledges the multifactorial burden hypertensive patients face, particularly in a post-pandemic Bangladeshi context. Our title clearly reflects this focus: “Quality of Life in hypertensive patients…”—emphasizing both the target population and the contextual setting.

Regarding novelty, our study adds unique value by: Applying WHOQOL-BREF among hypertensive patients in Bangladesh—a context where this instrument has been underutilized for this population; Capturing QOL after the COVID-19 pandemic, when hypertensive patients may have faced intensified health and psychosocial burdens; Identifying income, age, COVID-19 history, and lifestyle as significant determinants, which can guide locally tailored interventions. Our approach extends existing literature by using a culturally validated, multidimensional tool and by highlighting socioeconomic and pandemic-related contributors to poor QO

---

## [Editor Report · Decision Letter 1]

25 Jul 2025

Dear Dr. Akter,

Thank you for submitting your manuscript to PLOS ONE. After careful consideration, we feel that it has merit but does not fully meet PLOS ONE’s publication criteria as it currently stands. Therefore, we invite you to submit a revised version of the manuscript that addresses the points raised during the review process.

We look forward to receiving your revised manuscript.

Kind regards,

Md. Feroz Kabir, PhD, BPT, MPT, MPH, BPED, MPED

Academic Editor

PLOS ONE

Journal Requirements:

Additional Editor Comments:

Thanks for your response according to the reviewers comments. Please properly revise the English and format and submit within  Sep 08 2025 11:59PM.

---

## [Author Response · Author response to Decision Letter 2]

26 Jul 2025

Thank you for the revision request and considering our article for second revision. We are mentioning our revisions that made according to the recommendations. The point by point reply is mentioned below and revised manuscript is enclosed in the submission engine.

Journal Requirements:

Thank you for the suggestion. We carefully reviewed all prior queries recommended by the reviewer. Each of the suggested references was assessed for its relevance, methodological alignment, and contextual fit with the current study, which focuses on evaluating quality of life (QOL) among hypertensive patients in post-pandemic Bangladesh using the WHOQOL-BREF instrument.

• Relevant citations that aligned with our research objectives and added value to the background or discussion have been incorporated into the revised manuscript.

• Irrelevant or tangential references, those that did not directly support our study objectives, population, or methodological framework—were respectfully excluded, in accordance with the editorial guideline.

After a rigorous revision we updated our manuscript again. All incorporated references have been clearly cited in the appropriate sections of the manuscript and added to the reference list. We have ensured that only meaningful and scientifically relevant citations are included to maintain the coherence and focus of the manuscript.

Thank you for encouraging us to strengthen the manuscript through careful evaluation of the literature.

We sincerely appreciate the reviewer’s suggestion regarding the reference list. In response, we conducted a comprehensive review of all citations included in the manuscript titled “Quality of Life in Hypertensive Patients Using the WHOQOL-BREF Instrument in Post-Pandemic Bangladesh: A Cross-Sectional Study Followed Analytical Approach.”

• No retracted articles were identified among the cited references. We verified the status of each reference using reliable sources, including journal websites and the Retraction Watch database.

• All references have been reviewed and updated for accuracy and completeness, including author names, publication years, titles, journal names, volume/issues, and DOIs (where applicable).

• We also ensured that the referenced literature aligns with the scope, objectives, and context of our study, particularly regarding the use of WHOQOL-BREF, hypertension-related QOL assessments, and post-pandemic public health research in LMIC settings like Bangladesh.

The revised reference list reflects these updates, and the manuscript has been corrected accordingly. These changes have also been documented in the tracked version of the manuscript.

Thank you for your valuable feedback in ensuring the rigor and integrity of our work.

Additional Editor Comments:

Thanks for your response according to the reviewers comments. Please properly revise the English and format and submit within Sep 08 2025 11:59PM.

Thank you for your guidance. In response to your editorial note:

• We have conducted a thorough revision of the manuscript’s language to ensure clarity, grammatical accuracy, and professional tone throughout the text.

• The formatting has been carefully reviewed and corrected to align with the journal’s submission requirements, including headings, tables, figure legends, citations, and reference formatting.

• All changes have been tracked in the revised manuscript for transparency.

We appreciate your feedback and continued support throughout the review process.

---

## [Decision Letter · Decision Letter 2]

27 Nov 2025

Dear Dr. Nasrin Akter,

We look forward to receiving your revised manuscript.

Kind regards,

Kshitij Karki, MPH, MA

Academic Editor

PLOS ONE

Journal Requirements:

Additional Editor Comments (if provided):

Please revise as per the suggestions from the reviewers. Thank you

Reviewers' comments:

Reviewer's Responses to Questions

**Comments to the Author**

Reviewer #2: All comments have been addressed

Reviewer #5: All comments have been addressed

2. Is the manuscript technically sound, and do the data support the conclusions?

Reviewer #2: Yes

Reviewer #5: Yes

3. Has the statistical analysis been performed appropriately and rigorously?

Reviewer #2: Yes

Reviewer #5: Yes

4. Have the authors made all data underlying the findings in their manuscript fully available?

Reviewer #2: Yes

Reviewer #5: Yes

5. Is the manuscript presented in an intelligible fashion and written in standard English?

Reviewer #2: Yes

Reviewer #5: Yes

Reviewer #2: Thank you for addressing all comments in the revised manuscript titled "Quality of Life in hypertensive patients using the WHOQOL-BREF instrument in post pandemic Bangladesh: A cross-sectional study followed analytical approach". I have no further comments

Reviewer #5: Dr. Nasrin Akter and colleagues investigated Quality of Life in hypertensive patients using the WHOQOL-BREF instrument in Bangladesh. The major finding is that hypertensive patients have low QOL in psychological and social domains, with specific factors influencing QOL across all domains.

1. It is recommended that the title be revised to “Quality of Life in hypertensive patients using the WHOQOL-BREF instrument in the post-pandemic Bangladesh: A cross-sectional study”.

2. References 1 and 2 are listed in reverse order, as are References 24 and 25. Please revise these.

3. Line 212 states "Those with frequent no daytime napping were considered to have a poor sleep pattern." Are there any relevant references to support this argument?

4. As mentioned in Line 237, the "thirteen-item index" lacks one item in its parenthetical explanation. Please supplement it.

5. Table 1 and Table 2 are referenced in reverse order: Table 2 is mentioned in Line 279 while Table 1 is referenced in Line 297. Please correct these.

6. As referenced in Line 271, the value "n=269/300" corresponds to a calculated result of 89.7%. Please correct this accordingly.

7. In Table 1, the notation "*Scores < 1SD" appears, but the “*” is missing from the table. Please provide an explanation. Additionally, the position of "Spearman’s correlations (r)" is incorrect. Please adjust it accordingly.

8. It is more appropriate to revise Lines 305 and 306 to "with predominantly low to moderate relationships". Additionally, there is an error in the reference to Table 2 mentioned in Line 306 (before revision).

9. Could you please specify the details of the adjustments made to the model in the regression analysis mentioned in Line 333?

**Do you want your identity to be public for this peer review?** For information about this choice, including consent withdrawal, please see our Privacy Policy

Reviewer #2: No

Reviewer #5: No

---

## [Author Response · Author response to Decision Letter 3]

23 Dec 2025

Reviewer Query 1: It is recommended that the title be revised to “Quality of Life in hypertensive patients using the WHOQOL-BREF instrument in the post-pandemic Bangladesh: A cross-sectional study”.

Response 1: Thank you for the constructive suggestion regarding the title. We appreciate the reviewer’s recommendation and agree that the proposed title is clearer and more descriptive. Accordingly, we have revised the manuscript title to:

“Quality of Life in hypertensive patients using the WHOQOL-BREF instrument in post-pandemic Bangladesh: A cross-sectional study.”

This updated title more accurately reflects the study design, target population, and context.

Reviewer Query 2: References 1 and 2 are listed in reverse order, as are References 24 and 25. Please revise these.

Response 2: We revised the reference orders accordingly in the manuscript.

Reviewer Query 3: Line 212 states "Those with frequent no daytime napping were considered to have a poor sleep pattern." Are there any relevant references to support this argument?

Response 3: Thank you for pointing this out. The statement in Line 212 contained an unintentional typing error. Our intended definition was that frequent daytime napping, not the absence of napping, is considered an indicator of poor sleep patterns. This definition aligns with existing literature showing that individuals who nap frequently often do so due to inadequate or disturbed nighttime sleep, leading to overall poor sleep quality. Accordingly, we have revised the sentence to:

“Respondents reporting less than 6 (below standard) or more than 8 hours of sleep (Oversleeping), or those with self-reported insomnia or frequent no daytime napping, were considered to have a poor sleep pattern.”

A reference supporting this statement is added in the manuscript.

The correction has now been made in the manuscript.

Reviewer Query 4: As mentioned in Line 237, the "thirteen-item index" lacks one item in its parenthetical explanation. Please supplement it.

Response 4: 13-item index is enclosed as supplementary file.

Reviewer Query 5: Table 1 and Table 2 are referenced in reverse order: Table 2 is mentioned in Line 279 while Table 1 is referenced in Line 297. Please correct these.

Response 5: Thank you for your kind feedback. Actually, we rechecked the manuscript and confirmed that the citation of table 1 and table 2 were placed intentionally based on the flow and the contextual chronology in the text. Although the tables are mentioned in the reverse numerical order, each citation corresponds to the content being described at that specific point in the paragraph.

Reviewer Query 6: As referenced in Line 271, the value "n=269/300" corresponds to a calculated result of 89.7%. Please correct this accordingly.

Response 6: Thank you for your careful and critical observation. We corrected that accordingly.

Reviewer Query 7: In Table 1, the notation "*Scores < 1SD" appears, but the “*” is missing from the table. Please provide an explanation. Additionally, the position of "Spearman’s correlations (r)" is incorrect. Please adjust it accordingly

Response 7: Thank you for your careful review. In Table 1, the superscript notation “a” was intended to indicate that “poor scores” were defined as values <1 SD below the mean; however, due to an oversight, the footnote incorrectly displayed “*Scores <1 SD” instead of “aScores <1 SD”. We have corrected this by replacing the asterisk with the appropriate superscript “a” in the footnote so it now matches the table notation. Additionally, we have repositioned “Spearman’s correlations (r)” so that it appears directly above the correlation matrix for proper alignment. These corrections have now been incorporated into the revised table.

Reviewer Query 8: It is more appropriate to revise Lines 305 and 306 to "with predominantly low to moderate relationships". Additionally, there is an error in the reference to Table 2 mentioned in Line 306 (before revision).

Response 8: Thank you for your valuable insight. We addressed your concern and updated the manuscript accordingly in line 305.

Reviewer Query 9: Could you please specify the details of the adjustments made to the model in the regression analysis mentioned in Line 333?

Response 9: Thank you. We added a brief elaboration of model adjustment in lines 333-334 as-

“Model adjustment was performed using a backward elimination approach to control for potential confounders.”

---

## [Decision Letter · Decision Letter 3]

29 Dec 2025

Quality of Life in hypertensive patients using the WHOQOL-BREF instrument in post-pandemic Bangladesh: A cross-sectional study followed analytical approach

PONE-D-25-12356R3

Dear Dr. Nasrin Akter,

We’re pleased to inform you that your manuscript has been judged scientifically suitable for publication and will be formally accepted for publication once it meets all outstanding technical requirements.

Kind regards,

Kshitij Karki, MPH, MA

Academic Editor

PLOS One

Additional Editor Comments (optional):

Reviewers' comments:

Reviewer's Responses to Questions

**Comments to the Author**

Reviewer #5: All comments have been addressed

2. Is the manuscript technically sound, and do the data support the conclusions?

Reviewer #5: Yes

3. Has the statistical analysis been performed appropriately and rigorously?

Reviewer #5: Yes

4. Have the authors made all data underlying the findings in their manuscript fully available?

Reviewer #5: Yes

5. Is the manuscript presented in an intelligible fashion and written in standard English?

Reviewer #5: Yes

Reviewer #5: (No Response)

**Do you want your identity to be public for this peer review?** For information about this choice, including consent withdrawal, please see our Privacy Policy

Reviewer #5: No

---

## [Editor Report · Acceptance letter]

PONE-D-25-12356R3

PLOS One

Dear Dr. Akter,

I'm pleased to inform you that your manuscript has been deemed suitable for publication in PLOS One. Congratulations! Your manuscript is now being handed over to our production team.

Kind regards,

on behalf of

Dr. Kshitij Karki

Academic Editor

PLOS One